# Self-Efficacy in Online Teaching during the Immediate Transition from Conventional to Online Teaching in German and Argentinian Universities—The Relevance of Institutional Support and Individual Characteristics

Kerstin Göbel [1,*], Katharina Neuber [1], Carina Lion [2] and Uriel Cukierman [3]

1. Faculty of Educational Sciences, University of Duisburg-Essen, 45141 Essen, Germany
2. Department of Educational Sciences, School of Philosophy and Letters, University of Buenos Aires, Buenos Aires 1806, Argentina
3. Educational Research and Innovation Center, National Technological University, Buenos Aires 1190, Argentina
* Correspondence: kerstin.goebel@uni-due.de

**Abstract:** Triggered by the spread of the Coronavirus and the lockdown of universities in spring 2020, universities were required to provide infrastructure for digital teaching within a very short time. Further, all university members needed to develop knowledge and skills for teaching online. This paper presents data from the cross-cultural CRTS-Study (Coronavirus-Related Teaching Situation Study), which compares the experiences, attitudes and needs of university teachers in Germany and Argentina during the first lockdown in the context of the Coronavirus pandemic. The study has been carried out in spring 2020 as a cross-sectional online survey study with university teachers in Germany and Argentina ($N = 728$). The overall picture reveals a mostly successful implementation of online teaching for university teachers in both countries, with Argentinian university teachers reporting a slightly more positive perspective and slightly higher self-efficacy beliefs in online teaching when compared with the German colleagues. The results of regression analysis hint at the relevance of prior personal experience and institutional support for self-efficacy beliefs in online teaching for both samples. In conclusion, individual experience and training as well as supportive institutional conditions seem to be relevant for the development of digital teaching at universities in both countries.

**Keywords:** ERT; self-efficacy in online teaching; cross-country comparison; university teachers

## 1. Introduction

### 1.1. Online Teaching during the Beginning of the Coronavirus Pandemic

In December 2019, the SARS-CoV-2 virus was first discovered in Wuhan, the capital of Hubei Province, and has since then spread across national borders and continents. To limit the spread of the coronavirus, a lockdown of social life including universities and schools had been successively realized almost worldwide. In response to the first closure of the universities in spring 2020, university teachers ventured into often uncharted, unfamiliar terrain and designed an online teaching format for their courses overnight [1,2]. This crisis-induced transfer from face-to-face teaching to online-supported formats, named as "Emergency Remote Teaching (ERT)" [3], must be clearly distinguished from carefully planned digital teaching formats, as their primary aim was to counteract the inhibitory effects of the pandemic-related lockdown on the quality of higher education processes as quickly as possible and to enable alternative access to teaching and learning content and mentoring [4]. In the context of university lockdowns, the implementation of educational technology had been intensified to create a synchronous or asynchronous teaching offerings [5–8]. The switch to Emergency Remote Teaching (ERT) was commonly supported by

developing guidelines on the design of digital teaching and on the use of different digital tools, e.g., learning management platforms, videos, videoconference tools and others [9–15]. Still, neither the infrastructure of universities nor the competencies of university teachers were adequately prepared for this challenging transition from face-to-face to digital teaching and learning formats [16]. Nevertheless, the necessity to re-think and re-design higher education learning offered in the context of ERT was seen as an opportunity for stakeholders in higher education to reconsider the role of information and communication technologies (ICT), review its effectiveness, and hence increase resilience and sustainability of online learning in higher education for the future [17].

The ERT situation and the lockdowns due to the pandemic have been a challenge for university teachers and students [18]. During the first lockdown in spring 2020, students and university teachers used more educational technology tools than they were usually using before [19]. Most studies in the context of the switch to online teaching in higher education have been realized on student samples [20]. Students reported advantages with online learning, as they could listen to lectures from any places, which made online learning flexible, but due to few activating interactions and network instability, concentration in the learning process was partly reduced [21]. Further, students were concerned that during the pandemic their mental and physical health had deteriorated [22]; female students assessed a greater negative impacts of the pandemic, like social isolation, stress and mental health problems, compared with their male counterparts [23]. Nevertheless, female students reported being more active in learning and their satisfaction with processes of online learning and the university was higher than that of male students [24].

Compared with the various experiences of students, the experience of university teachers during the pandemic has only been addressed in a few studies [5,25–32]. Results indicate that university teachers experienced the new teaching and learning situation as largely successful, rating it more positively may have been expected [5,25–28]. Several potentials for the switch to online teaching have been perceived by university teachers, e.g., more flexibility and autonomy in learning as well as digital competence development [5,29–32]. Nevertheless, individual characteristics of university teachers with regards to digital technology, such as former experience, a positive attitude and self-efficacy beliefs seem to play a relevant role for successfully managing the rapid shift to online teaching [6,33,34].

### 1.2. The Relevance of Self-Efficacy Beliefs for Online Teaching

Aspects of self-efficacy beliefs might be important for the realization of online teaching [35], and self-efficacy beliefs may even function independently of underlying skills [36]. Self-efficacy beliefs can be conceptualized in a quite general manner, but also in a more specific manner depending on the context, the environment, and the specific task [37]. From a general learning perspective, self-efficacy beliefs are nurtured from different sources, but experiences of mastery seem to be most important [38]. Concerning expertise in face-to-face teaching, there is significant evidence for the correlation between teachers' self-efficacy beliefs, their teaching performance and students' learning success in schools [39,40]. However, as online teaching is different from face-to-face teaching, a specific examination of self-efficacy in online teaching seems necessary. From research in the context of the technology acceptance model, we know that self-efficacy beliefs are related to perceived usefulness and ease of use of digital tools [41]. More specifically, technology enhanced self-efficacy beliefs of students are shown to be associated with a higher perception of ease of use; hence result in a higher willingness to use technology [42–44]. Another recent study on self-efficacy beliefs in online learning indicates a negative correlation between self-efficacy beliefs and difficulties in online learning for students [45]. Concerning the use of digital technology, motivational aspects seem to be especially relevant for female students [44].

Consequently, teachers' self-efficacy beliefs regarding digital teaching and digital teaching tools should also be influential for the implementation of online teaching formats [46]. When trying to explain perceived effectiveness and skills in dealing with technologies in higher education, empirical findings show that the actual use of digital tools [46,47], and

the perceived usefulness of these technologies [48] play a relevant role. However, when looking at the literature concerning online teaching, there are still only a few studies address self-efficacy beliefs and teaching quality in online teaching contexts from the perspective of university teachers [49].

In comparison, a large body of studies describe conditions and challenges in online teaching, aiming at differentiating different competencies and levels of university teachers' expertise in online teaching [50–53].

In the context of a study with Chinese university teachers, most respondents lacked experience in online teaching at the beginning of the first lockdown, but technology application increased during online teaching, and although general self-efficacy beliefs in online teaching had not been perceived to have increased, self-efficacy in online applications still increased among Chinese university teachers during the lockdown [54]. From the perspective of university teachers, the increase in online teaching goes along with increased flexibility and more independent student learning, which is evaluated positively. However, university teachers point at the problem of maintaining the relationship with students; the lack of contact is a relevant problem which should somehow be addressed in adequate online formats [54,55]. A survey study in the United States realized during ERT reveals a relevant shift in online teaching strategies towards a more instructor-centered mode in online teaching, which was more detached from students when compared with face-to-face teaching [56]. Scherer and Colleagues assume that university teachers' mastery of online teaching might be dependent on individual and contextual variables, arguing that individual variables like gender and online teaching experience as well as contextual variables of the institution might play a role for the success of online teaching in the context of the pandemic related ERT [57].

Results from an international study on university teachers suggest that although the quality of teaching was impeded, university teachers tried to maintain teaching quality despite the difficult situation, but the actual success in maintaining teaching quality seems to be highly determined by prior personal experience [58]. Dorfsman and Horenczyk also point to the relevance of digital literacy before the pandemic related ERT as predictor for mastery in online teaching [50]. Besides individual experience in online teaching, individual characteristics might also be relevant for dealing with online teaching formats. Following the idea of a global digital gender divide in the use of digital technology [59,60], gender-specific attitudes and experiences might have an influence on individual openness to digital teaching and learning formats, but the results concerning university teachers seem heterogeneous. In a study with Spanish university teachers investigating their attitudes towards ICT (information and communication technologies), the authors found that female university teachers tend to report lower general positive attitudes towards ICT than their male peers [60]. A similar picture shows up in an international teacher survey, where female teachers seem to be less engaged in digital teaching than their male peers [61]. In contrast, a study in the US revealed a higher self-efficacy level in online teaching instruction for female university teachers [62]; furthermore, the perception of student learning was highly associated with the self-efficacy beliefs of the university teachers in this survey.

While it can be stated that prior personal experience with digital technology and online teaching of university teachers is influential for self-efficacy beliefs in digital teaching, the context where digital teaching is implemented seems a further relevant framing condition of individual performance and the development of expertise. Several studies hint at the relevance of an enabling environment, which integrates and supports the use of digital technology on campus as being influential for the individual mastery of online teaching [57,63]. A common concept of the learning process seems helpful for the use of digital media in times of ERT at universities [63]. Further, a study on college teachers in Indonesia during ERT could confirm that perceived organizational support had a significant influence on university teachers' online teaching self-efficacy and on their readiness for change [64]. In contrast, a study on American university teachers from the nursing faculty revealed that

their online teaching self-efficacy could be predicted by prior online teaching; institutional support did not appear to be predictive for perceived online teaching self-efficacy [65].

Summing up, empirical studies relating to the mastery of online teaching during ERT hint at the relevance of individual characteristics, such as prior experience, on the one hand, and institutional support aspects on the other hand.

## 2. Comparative Perspectives on ERT

Although there is a wide range of research on the differences made by ERT, comparative approaches to understand possible differences between cultures and populations are still scarce [6]. Studies prior to ERT hint at the relevance of cultural influences on technology use and acceptance in educational settings, which has been discussed in the context of levels of technological development in the countries studied [66]. A comparison between German and US university teachers revealed differences in self-efficacy beliefs concerning online teaching, where US teachers reported higher levels of self-efficacy and relevance of the integration of educational technology in their teaching [51]. The differences might be interpreted as German university teachers having less experience in the use of educational technology or may also be attributed to different perceptions of the constructs in use [67]. In the context of ERT, studies show that measures have been taken to deal with this situation in different contexts, but the institutional preconditions concerning digital teaching still vary between countries, which might have led to different measures [4,8].

The CRTS-Study (Coronavirus-Related Teaching Situation Study) has been conceptualized at the beginning of ERT by researchers from Israel, Argentina, Switzerland, France, UK, and Germany, which allows for comparative perspectives on the implementation of online teaching in ERT [5,6]. In the context of a comparison between European countries (Germany, Switzerland, France, and UK), Germany and Switzerland have quite high self-efficacy perceptions concerning online teaching. All university teachers reported a higher use of educational technology, especially concerning synchronous web conferencing systems, during the first lockdown than before, which implies that they adapted their teaching to the ERT situation [6]. As the study of Kaqinari and colleagues focused on European university teachers, the present article expands the focus by comparing university teachers from Argentina with university teachers from Germany, looking at their perceptions of the transition process and analyzing the relevance of individual and institutional factors for their self-efficacy beliefs. The ERT situation in both countries is described briefly in the following section.

### 2.1. Online Teaching at German Universities

In spring 2020, the coronavirus spread rapidly and extensively in Germany and all over the world. While the number of laboratory-confirmed infections with the coronavirus had doubled in Germany, the federal states ordered closures of public educational institutions, such as universities, schools, and childcare centers. On 16 March, Germany implemented a widespread lockdown and enacted various arrangements to slow down the spread of the coronavirus. The lockdown included restrictions on public life (e.g., by closing restaurants, bars, stores, and entertainment and recreational facilities) and was meant to minimize social contact [68,69]. 'Social distancing' has since been considered as one of the most important guidelines in the fight against the coronavirus [70]. Facility closures and extended contact restrictions resulted in many businesses going to part-time work or employees working in home offices. The guidelines on restrictions on public life as well as social distancing could help to reduce the number of new daily coronavirus infections in Germany. A gradual lifting of restrictions was announced in Germany on 15 April [71]. Still subject to compliance with special hygiene guidelines (use of masks and contact tracing), starting in May, schools, and even libraries and stores were allowed to gradually reopen.

In compliance with the respectively valid legal situation, universities in Germany, Austria and Switzerland planned different online study formats to avoid a "lost semester" for students. All university teachers had to prepare to realize technology-mediated teaching

and learning formats almost immediately [72]. The digitization of teaching and learning at universities in Germany was already being demanded before the outbreak of the coronavirus pandemic [73] and had gained interest at the level of higher education management [74–76]. The use of digital media was estimated to have a potential to enrich existing learning opportunities and materials in addition to face-to-face teaching. Furthermore, by providing asynchronous formats, such as videos or recorded presentations, the diversity of students' needs might be better met, as they can be used independently of time and location [77]. Hence, the use of digital tools in higher education was seen to promote individualized and flexible learning experiences, and potentially enhance the didactic quality of teaching and the acquisition of competencies by students [74,78,79]. In the years before the coronavirus-related lockdown, Riedel carried out a study with university teachers in Germany concerning their digital teaching [80]. The majority could be characterized as 'material users', of whom about half of all respondents could be counted. This group only integrated individual digital learning materials, such as texts or videos, into their teaching. Approximately 30% of the respondents could be assigned to the group of 'multimedia users', using tools that enabled digital presentations and video conferencing with above-average frequency. Only about 18% of the respondents declared using digital tools intensively. Many university teachers reported that they did not have previous experience with digital teaching prior to the first coronavirus-related ERT (digital summer semester 2020) [81]. Birkenrahe, Hingst and Mey also address the issue of insufficient experience among university teachers, pointing to reasons such as a lack of media competence and having too little time to improve upon this [82].

A German survey of professors and students In 2020 showed that, overall, the German universities have coped well with the challenges due to the switch to online teaching caused by the coronavirus pandemic [83,84]. Teaching was largely maintained in the coronavirus-impacted teaching semesters and according to the interviewed professors, only a few lectures and seminars had to be cancelled without replacement, and it was still possible to take examinations [83]. A German online survey in 2020 with around 25,000 students [84] revealed that, regardless of whether the respondents were freshmen or not, digital teaching was viewed ambivalently; on the one hand, students appreciated the time flexibility that digital teaching formats allow; on the other hand, students missed the contact with fellow students and university teachers. Preparing for exams and taking them digitally is also viewed rather critically. Most students' computers allowed them to participate in digital teaching formats without any problems. However, the capacity of the internet connection at home was not always sufficient. The living situation was not perceived to be ideal for digital teaching and studying for all students, hence, many students feared that their study time will be extended due to the pandemic [84,85].

### 2.2. Online Teaching at Argentinian Universities

Once the pandemic was declared by the World Health Organization (WHO) in March 2020 [86], Argentina began preparing for the response through timely detection of sick people arriving in the country in order to contain the disease and mitigate its spread. Among these measures, the preventive and compulsory social isolation (ASPO, according to the Spanish acronym) stands out for those who do not work in essential sectors of the economy throughout the country, which came into force early on 20 March [87].

Due to ASPO, all schools and universities in Argentina closed their buildings and transformed their regular activities into a fully virtual mode in just a few weeks. The universities made efforts to continue teaching within the framework of educational policies. The students have been able to continue their educational activities beyond the emergency. The university system promoted conditions of equal opportunity of access to technological resources in the development of the virtual modality, by means of scholarships, connectivity agreements with service providers, making course regimes more flexible, implementing tutorial accompaniments, and materially assisting those in need. This virtual-learning modality arose as a precedent for the future, both in virtual work linked to various aspects

of institutional management and in access to higher education through virtual platforms, demonstrating the universities' capacity, commitment, and quality in guaranteeing the continuity of studies [88]. Several programs were developed by the Ministry of Education to support transition, initially to the virtual modality and later to the hybrid modality [89–91]. The general scenario was that the academic community was unprepared but still able to deliver. Nevertheless, inequalities became more visible and issues started to emerge in the debates in the government [92].

The context of COVID-19 has deeply penetrated various aspects of university life, such as academic and administrative management; teaching practices and learning have been challenged and, in some way, transformed. During the second half of the year 2020, the Secretary of University Policies conducted a series of surveys at different university levels: authorities, professors, students and non-teaching workers to generate systematic data on the effects of COVID-19 in the organization of academic, work and family life within the university community [89]. The results of this survey showed that almost all universities (99.5%) decided to transform their courses into a virtual modality and 87% of them were effectively developed according to the proposed objectives. The reasons for not being able to make this transformation varied, such as a lack of technological resources, difficulties teaching in a virtual mode, or not having enough time to reorganize the course. It is important to point out that more than 60% of the professors said that they did not have previous experience in distance education before the suspension of face-to-face lectures. The most widely used technological tools during this period were learning management systems, e-mail, and videoconferencing. Some professors said that they also used instant messaging tools. When asked about the percentage of the course content covered under the virtual mode, 61% answered that they were able to cover more than 80%, and only 24% developed between 60% and 80% of the content. Interestingly, almost all professors (96.4%) declared that they were able to evaluate students. There was no major agreement about the questions related with the training provided by the university regarding the technological and computer resources necessary for lecturing in the virtual mode. Finally, more than 80% of the professors said that they were satisfied in general with the development of their courses and 67% said that they were able to complete their courses.

Two of the biggest universities in Argentina conducted a survey among 400 university teachers asking about the changes brought about by the pandemic to higher education [27]. Results from the survey reveal that the use of technologies increased compared with the usage before the coronavirus pandemic. The authors conclude that the context forced the use of synchronous tools and virtual environments. Furthermore, it could be shown that the support of the university was important for the feasibility of good practices through teacher training. In those cases where faculties were able to access tools and training, teachers recognized that they were able to carry out their teaching in a better way. Moreover, the results show that there was a high degree of adaptability among university teachers despite not having chosen this modality. In addition, they found that many teachers consider that they have managed to improve their lessons with the inclusion of technologies and generated other bonds with their students. Finally, there is a recognition of diverse good practices according to each professional field. In the case of academic fields that require a high load of practical teaching, working online has been more complex.

### 2.3. The Present Study

The present article focuses on university teachers' perspectives on the implementation of online teaching and the associated challenges in times of the first coronavirus-related lockdown of the universities. As comparisons between different contexts concerning the perception of ERT and the concept of self-efficacy are still scarce, the present article focuses on the comparison of the perception of the transition to online teaching in the first lockdown and at the relevance of individual and institutional predictors for the perception of self-efficacy beliefs in online teaching, comparing the perspectives of university teachers from German and Argentinian universities.

The paper presents exploratory results regarding the following research questions:

(1) How did German and Argentinian university teachers experience the transition from face-to-face to online teaching?

(2) How do German and Argentinian university teachers assess the success of their first online teaching experience, and do they differ in their self-efficacy beliefs in online teaching?

(3) To what extent do personal characteristics, individual competency, and relevant institutional factors correlate with the perception of self-efficacy beliefs in online teaching in both countries?

## 3. Method

### 3.1. Study Design

The present data is derived from a larger study context, which is the CRTS study (Coronavirus-Related Teaching Situation Study). It aims to investigate how university teachers experienced the challenging situation of the immediate transition from face-to-face to online teaching in the initial coronavirus-related lockdown in Spring 2020. The online survey was based on a questionnaire approved by the Ethics Committee of the Faculty of Education at the Hebrew University. The questionnaire (see Appendix A) was developed jointly by the teams participating in the CRTS project (Initiators of this study are: Prof. G. Horenczyk and Dr. M. Dorfsman (Hebrew University, Israel); Dr. C. Lion (University of Buenos Aires, Argentina); Prof. K. Göbel (University of Duisburg-Essen, Germany); Prof. E. Makarova (University of Basel, Switzerland); Dr. D. Birman (Miami University, USA).) covering the following topics: pedagogical–didactic challenges and the ways in which university teachers deal with these challenges; needs and attitudes related to the transition of teaching; and the extent to which the university responds to the challenges and needs of academic staff according to the university teachers' assessment.

The present paper is based on an online survey with university teachers from German universities (headed by researchers from the University of Duisburg-Essen) and university teachers from Argentina (headed by researchers from the University of Buenos Aires and the National Technological University). The participants were surveyed with an online questionnaire focusing on attitudes towards the transition to online teaching, self-assessed competency for online teaching, use of digital tools before and during the lockdown, evaluation of the preparatory process and evaluation of online teaching units. Furthermore, age and gender of participants were assessed.

### 3.2. Participants

A total of $n = 292$ university teachers from German universities (176 of them female; 63.1%) and $n = 436$ university teachers from Argentina (209 of them female; 48.4%) took part in the online survey on the teaching situation in the time of the coronavirus pandemic. The German sample consists of university teachers from eleven different universities, with most participants belonging to the University of Duisburg-Essen ($n = 154$, 86.5%). The Argentinian sample equally includes participants from the University of Buenos Aires (UBA, $n = 219$) and from the National Technological University (UTN, $n = 217$), with most university teachers currently teaching in the University of Buenos Aires ($n = 102$, 23.4%), followed by regional faculty of General Pacheco of UTN ($n = 44$, 10.1%).

The difference in the gender distribution between both countries was found to be significant (chi-square (1, $n = 711$) = 14.760, $p < 0.001$). Regarding age and teaching experience in the tertiary sector, a heterogeneous composition of the overall sample emerges (see Table 1). An age range of 26 to 35 years is most frequently reported by respondents from the German universities (31.4%), while in the Argentinian sample most university teachers reported an age range of 46 to 55 years (35.8%). Overall, participants from German universities appear to be younger than university teachers from Argentina (see Table 1): 58.9% of respondents from Germany indicate an age below 45 years; this applies to only a third of the respondents from Argentina (27.3%). Concerning the teaching experience in

the higher education sector, a similarly different distribution of answers emerges among university teachers from both countries (see Table 1).

**Table 1.** Valid and cumulative percentages to describe the sample.

| | German University Teachers | | | Argentinian University Teachers | | |
|---|---|---|---|---|---|---|
| | **N** | **%** | **Cum. %** | **N** | **%** | **Cum. %** |
| | | | Gender | | | |
| female | 176 | 63.1 | 63.1 | 209 | 48.4 | 48.4 |
| male | 193 | 36.9 | 100 | 223 | 51.6 | |
| Total | 279 | 100 | | 432 | 100 | |
| | | | Age in years | | | |
| 25 and younger | 1 | 0.3 | 0.3 | 1 | 0.2 | 0.2 |
| 26–35 | 90 | 31.4 | 31.7 | 49 | 11.2 | 11.5 |
| 36–45 | 78 | 27.2 | 58.9 | 69 | 15.8 | 27.3 |
| 46–55 | 61 | 21.3 | 80.1 | 156 | 35.8 | 63.1 |
| 56–65 | 50 | 17.4 | 97.6 | 138 | 31.7 | 94.7 |
| older than 65 | 7 | 2.4 | 100 | 23 | 5.3 | 100 |
| Total | 287 | 100 | | 436 | 100 | |
| | | | Teaching experience in the tertiary sector in years | | | |
| 1–5 | 91 | 31.6 | 31.6 | 25 | 5.7 | 5.7 |
| 6–11 | 68 | 23.6 | 55.2 | 66 | 15.2 | 20.9 |
| 12–17 | 55 | 19.1 | 74.3 | 76 | 17.5 | 38.4 |
| 18 and more | 74 | 25.7 | 100 | 268 | 61.6 | 100 |
| Total | 288 | 100 | | 435 | 100 | |
| | | | Teaching hours per week | | | |
| 1–2 | 66 | 23.0 | 23.0 | 12 | 2.8 | 2.8 |
| 3–6 | 93 | 32.4 | 55.4 | 95 | 21.8 | 24.5 |
| 7–11 | 83 | 28.9 | 84.3 | 173 | 39.7 | 64.2 |
| 12 and more | 45 | 15.7 | 100 | 156 | 35.8 | 100 |
| Total | 287 | 100 | | 436 | 100 | |

Regarding the average teaching time per week, there are differences between the study groups (see Table 1). While at the German universities more than half of the university teachers surveyed stated that they teach between one and six hours per week (55.4%), this only applies to 24.5% of the respondents at the Argentinian universities, where 35.8% of the respondents reported to teach more than 12 h per week; this is true for only 15.7% of German university teachers. Likewise, the samples significantly differ regarding the received support (e.g., student assistant) in the preparation and implementation of teaching activities (chi-square (2, $n = 718$) = 207.782, $p < 0.001$). While more than half of the respondents in Argentina ($n = 235$, 53.9%) state that they receive support in all courses, this is only true for 23 respondents from Germany (8.2%). The majority of German respondents state that they do not receive any support ($n = 211$, 74.8%).

*3.3. Measure*

This paper focuses on university teachers' assessments of the pedagogical–didactic challenges during the coronavirus-related teaching situation at their university and on the university teachers' self-efficacy beliefs concerning online teaching.

To better understand how university teachers perceive the pedagogical–didactic challenges during the coronavirus-related teaching situation at their university, the university teachers were asked to indicate the extent to which they had used or were using vari-

ous digital tools before and during the lockdown of the universities (1 = "not at al" to 4 = "to a large extent"). Retrospectively, the university teachers were asked to describe their experience with the transition from conventional to online teaching (ranging from "very positive and inspiring", via "mostly positive and reassuring" to "complex", "frustrating" or "nothing special") as well as the implementation of the first online-based teaching units (ranging from 1 = "unsuccessful" to 5 = ""very successful""). Furthermore, university teachers were asked to articulate support needs (To what extent would you expect support of your institution in time of a future crisis?). The assessment of support needs in future crisis situations (time for preparation, individual support, and monetary compensation) was based on a four-point scale from 1 = "not at all" to 4 = "to a large extent".

It was of special interest to understand how the university teachers assess their abilities to implement online teaching using digital tools. Self-assessed ability to implement online teaching was captured via a self-efficacy beliefs scale, which included questions about the extent to which university teachers are confident in their ability to successfully teach online during university closures. In total, eight items from two existing scales [93,94] were adopted and modified for the coronavirus-related teaching situation (for example: I feel confident that I am able to select the most efficient digital tools for teaching in this situation). All items were answered with a Likert scale from 1 = "not at all" to 4 = "completely agree". Analyses revealed satisfactory internal consistencies for both samples (Germany: Cronbach's $\alpha = 0.83$, Argentina: Cronbach's $\alpha = 0.79$).

Retrospectively, the participants were asked about the reasons for successful online teaching in the current context; we distinguish between internal, personal reasons, such as own technological skills or the competence of addressing issues faced by the students; and external, context-related conditions, such as institutional support or sense of emergency. The response options ranged from 1 = "not at all" to 4 = "to a large extent". Three items considered institutional support factors: technological pedagogical support of the academic institution; existing online tutorials; support and encouragement of administration/senior management of the institution; these three items were combined into one scale for further regression analysis (Germany: Cronbach's $\alpha = 0.67$, Argentina: Cronbach's $\alpha = 0.69$).

Finally, personal characteristics concerning gender, age and occupational experience in years were surveyed.

*3.4. Analysis*

The data collected on the experiences and assessments of the coronavirus-related teaching situation were analyzed descriptively at the level of the individual items. Non-parametric procedures (chi-square and Mann–Whitney U tests; fixed significance level of 5%) were calculated to determine if there were differences between university teachers from Germany and Argentina.

To examine the extent to which personal characteristics, own competencies and support needs influence the university teachers' self-efficacy beliefs in online teaching, the data collected were analyzed using multiple regression models. In an overall regression model, the country emerged as a relevant predictor of self-efficacy beliefs, so we decided to analyze the regression models for each country separately and compare the model results descriptively with each other. Stepwise regression equations were carried out to identify the respective explanatory power of the resulting models. The first model only includes personal characteristics (gender; age; occupational experience in years); in the second model, reasons for successful online teaching were integrated as dummy variables (Coding: 0 = not at all/small extent; 1 = moderate/large extent); three items considered institutional support factors were combined into the scale institutional support factors (see above). The final model also contains expected support needs in further crisis situations (Coding: 0 = not at all/small extent; 1 = moderate/large extent).

As former studies hint at the relevance of prior experience and competence concerning the use of digital tools for the mastery of online teaching [6,47,55], prior experience might also be a relevant predictor for the perception of self-efficacy. In our study, we measured

prior and actual online teaching experience with the scales "use of digital tools before lockdown" (Germany: $\alpha$ = 0.53; Argentina: $\alpha$ = 0.64) and "use of digital tools during lockdown" (Germany: $\alpha$ = 0.27; Argentina: $\alpha$ = 0.47). The scales were developed by averaging the respective items. We decided not to integrate the corresponding scale "use of digital tools before lockdown" into our regression models due to its insufficient internal consistency. However, exploratory correlation analyses were conducted to investigate the potential relationship between self-efficacy beliefs in online teaching and the use of digital tools.

## 4. Results

### 4.1. Experience with the Transition to Online Teaching

Results from the German and Argentinian survey concerning the use of digital tools reveal that university teachers from both universities used LMS platforms and presentations (e.g., PowerPoint) to a moderate or large extent in their own teaching, both before and during the closure of the universities (see Figure 1). The comparison of the countries reveals that respondents from Argentina used digital media more frequently as part of their conventional teaching than university teachers from Germany before the outbreak of the coronavirus pandemic; in particular, LMS platforms for bibliography ($U$ = 51,354.500, $Z$ = −4.343, $p$ < 0.001), discussion forums ($U$ = 35,829.000, $Z$ = −8.826, $p$ < 0.001), selected videos ($U$ = 45,014.000, $Z$ = −6.112, $p$ < 0.001), self-produced videos ($U$ = 34,574.500, $Z$ = −10.393, $p$ < 0.001), and online lessons via Zoom or other ($U$ = 28,191.000, $Z$ = −13.447, $p$ < 0.001) were used significantly more frequently by university teachers from Argentina. During the closure of the universities (lockdown), especially web conference systems and LMS platforms were used increasingly in university teaching in Germany as well as in Argentina to set up digital discussions and group work (see Figure 1). The most striking increase in usage was experienced in online teaching via Zoom or other web conferencing systems, where their functions allow synchronous distance teaching. Across all digital media, there was an increase in use in both Germany (before: $M$ = 2.28, $SD$ = 0.48; during: $M$ = 2.95, $SD$ = 0.49; $\Delta_M$ = 0.678, $t$(288) = 23.287, $p$ < 0.001, Cohen's $d$ = 1.370) and Argentina (before: $M$ = 2.79, $SD$ = 0.64; during: $M$ = 3.22, $SD$ = 0.53; $\Delta_M$ = 0.429, $t$(433) = 17.007, $p$ < 0.001, Cohen's $d$ = 0.816). University teachers from Argentina show a higher extent of digital tool use in general (before: $t$(711.214) = −12.337, $p$ < 0.001, Cohen's $d$ = −0.885; during: $t$(721) = −6.893, $p$ < 0.001, Cohen's $d$ = −0.523), both before and during the lockdown.

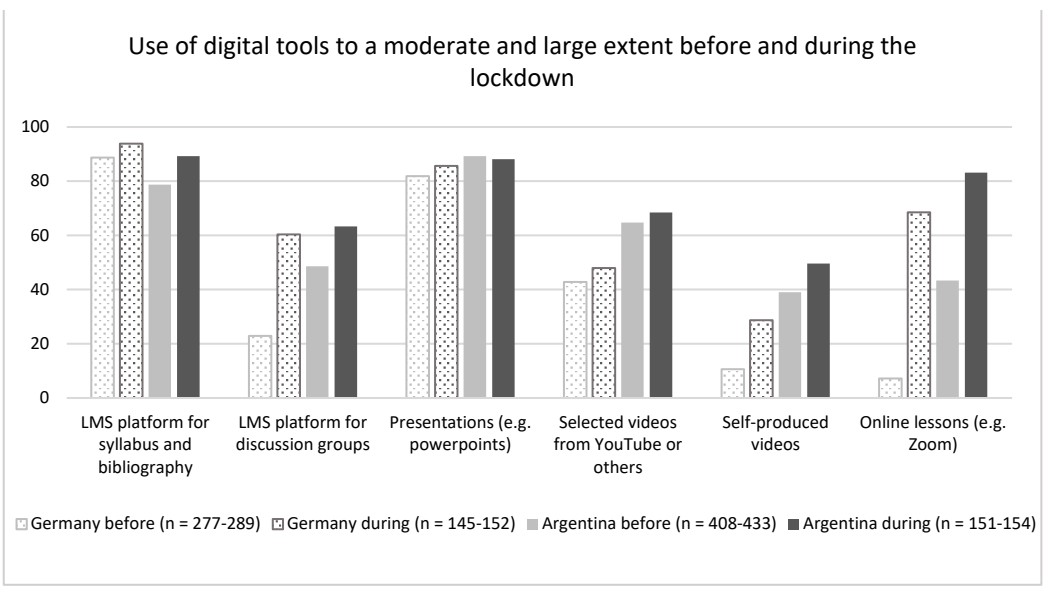

**Figure 1.** Use of digital tools to a moderate and large extent before and during the lockdown. Valid percentages indicated (agreement).

Overall, the experience of the switch to online teaching is perceived ambivalently by the respondents (see Figure 2). In both groups, the experience is perceived as a positive experience, while university teachers from Argentina rate it even more positively than the German colleagues. Both samples differ significantly in their perceptions of the transition to online teaching as 'a very positive and inspiring experience' ($U = 35,970.000$, $Z = -12.266$, $p < 0.001$) and as 'a mostly positive and reassuring experience' ($U = 46,938.000$, $Z = -7.330$, $p < 0.001$). In the German sample, the greatest agreement to the question "How would you describe your experience during the coronavirus-related teaching situation (CRTS)" is found for the statement that the transition of teaching was 'a mostly positive and reassuring experience' (49.7%); in the Argentinian sample, the greatest agreement was found for the statement that the transition to online teaching was 'a complex experience; it requires investment beyond what is expected' (57.3%). The teachers at German universities agreed with this statement (33.9%) significantly less often ($U = 48,738.000$, $Z = -6.199$, $p < 0.001$). On the other hand, the proportion of university teachers who perceived the transition to online teaching as 'a frustrating experience' was significantly higher at German universities than at the universities in Argentina (Germany: 13.4%; Argentina: 3.2%; $U = 57,198.000$, $Z = -5.160$, $p < 0.001$). For 12.7% of the respondents from Germany and 2.3% of the respondents from Argentina, the transition to online teaching was 'nothing special' ($U = 67,050.000$, $Z = -5.581$, $p < 0.001$).

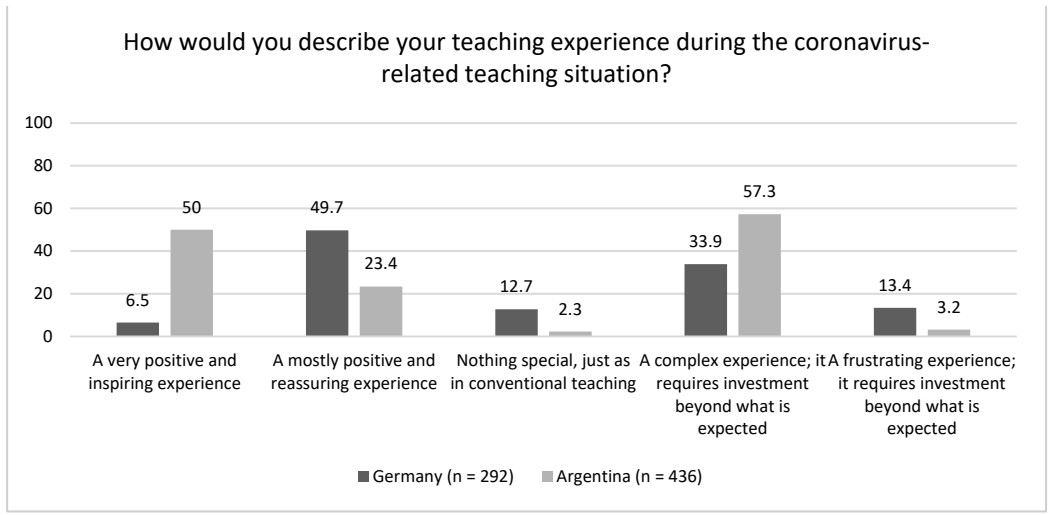

**Figure 2.** Perceptions of the experience of the transition to online teaching. Valid percentages indicated (agreement); significant differences between the samples were found for all items.

Retrospectively, the university teachers were asked about the need for support concerning online teaching in possible crisis situations in the future. In the event of a similar crisis in the future, most respondents in Germany (70.8%, $n = 197$) and Argentina (91.5%, $n = 369$) expect more time and resources to be able to prepare for the transition of teaching (agreement to a moderate/large extent). Furthermore, university teachers expect individual support from experts in educational technologies or instructional design to assist during online teaching (Germany: 68.4%, $n = 189$; Argentina: 84.4%, $n = 346$). Finally, a majority of university teachers in both countries have an expectation of receiving monetary compensation over and above their salaries for preparing to teach online (Germany: 57.3%, $n = 155$; Argentina: 57.3%, $n = 223$).

### 4.2. Success of Online Teaching and Self-Efficacy Beliefs

Although the experiences with online teaching differ slightly between university teachers from Germany and Argentina (see Figure 2), the preparation process for the immediate transition of teaching is rated similarly overall by university teachers from both countries, and most of them did not perceive it as a major difficulty. In both countries, the great-

est agreement was found for the statement that the preparation process was reasonable (Germany: 37.6%; Argentina: 45.1%). Furthermore, 42.9% of German participants and 41.3% of respondents from Argentina rated the preparation for online teaching as simple or very simple. In both samples, only a small number of university teachers found the preparation for online teaching difficult or very difficult (Germany: 12.4%; Argentina: 5.3%). In line with this, the first lessons with Zoom or other web conferencing systems were predominantly assessed positively by the university teachers surveyed. Slightly less than half of the university teachers from both German universities (40.4%) and Argentinian universities (40.3%) assessed their first online teaching during the lockdown as very successful or successful. More than one-third of the respondents from both countries rated the implementation of their first online lessons as reasonable. While 21.9% of the respondents at German universities felt that there is room for improvement, this is true for only 10.6% of the respondents at Argentinian universities.

The university teachers attributed the success of their online teaching mainly to internal, personal aspects, such as own technological skills and the competency for addressing students' problems (see Figure 3). However, the groups differ significantly from each other in the perceived extent of different reasons' relevancy to their success in online teaching. In the German sample, the statement that the success of teaching was determined by the need to do it ad hoc (external condition: sense of emergency) was rated to a higher extent than in the Argentinian sample. Less important for the success of online teaching during the lockdown were external factors of institutional support (e.g., technological pedagogical support of the academic institution and the availability of tutorials). German participants in particular attributed their teaching success to institutional support only to a small extent; the agreement is significantly lower than in the Argentinian sample ($t(697) = -11.148$, $p < 0.001$, Cohen's $d = -0.860$).

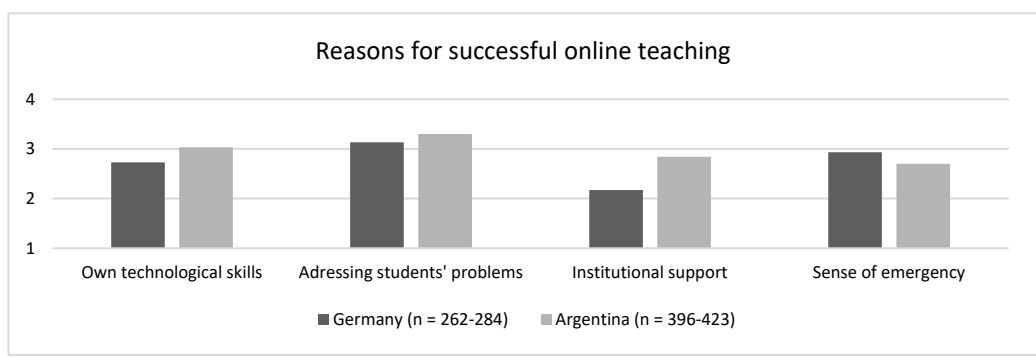

**Figure 3.** Perceptions of reasons for successful online teaching. Mean values based on a scale from 1 (not at all) to 4 (to a large extent); significant differences between the samples were found for all items.

University teachers' beliefs about their competencies in digital teaching were captured using a self-efficacy expectancy scale (see method section), which included questions about the extent to which university teachers are confident in their ability to successfully teach online during university closures (see Figure 4). Overall, the results indicate that there is a high self-efficacy expectation for online teaching among the university teachers, and they present themselves as being confident to teach successfully even under the more difficult conditions. As we can see from the mean values for the items (see Figure 4), the Argentinian university teachers rate their self-efficacy concerning online teaching even higher than the German respondents. Accordingly, we found a significant difference between the mean scores of Argentinian university teachers ($M = 3.37$; $SD = 0.45$) and the German respondents for the self-efficacy scale ($M = 3.06$; $SD = 0.48$; $t(720) = -8.809$, $p < 0.001$, Cohen's $d = -0.669$).

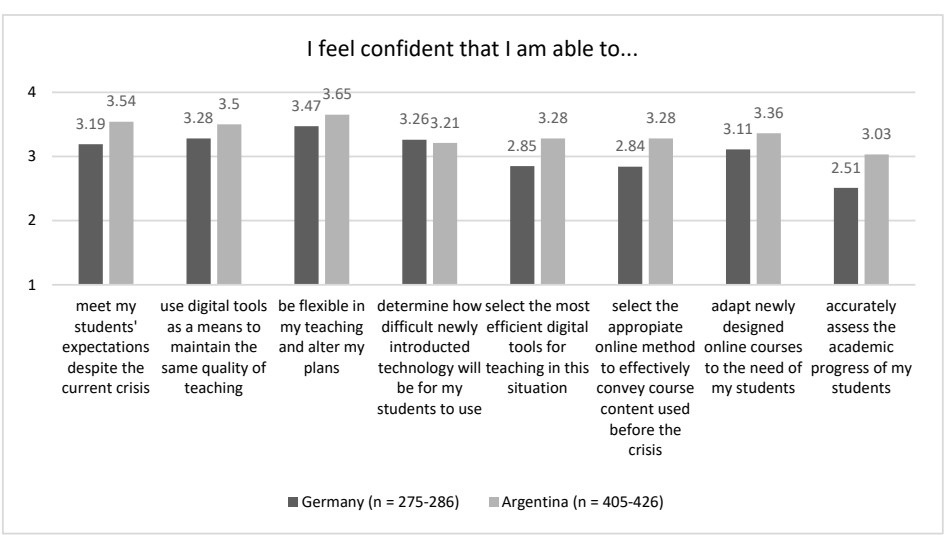

**Figure 4.** Self-efficacy beliefs concerning online teaching. Mean values based on a scale from 1 = "not at all" to 4 = "completely agree".

Further explorative analyses indicate a correlation between self-assessed competency in online teaching and the use of digital tools. Overall, it can be stated that the more self-efficacious the university teachers rate themselves, the more extensively do they use digital tools for online teaching. The correlation between the extent of digital tool usage before as well as during the lockdown and the self-assessed ability to teach online proved to be low, but statistically significant for both the German sample (before: $r = 0.243$; during: $r = 0.239$; $p < 0.001$) and the Argentinian sample (before: $r = 0.216$; during: $r = 0.227$; $p < 0.001$).

*4.3. Predictors of the Perception of Self-Efficacy in Online Teaching*

To examine the relevance of personal characteristics, experience, and support needs to the perceived self-efficacy in online teaching, a multiple stepwise regression was conducted for the German as well as for the Argentinian data.

The first model included only personal characteristics (Germany: $F(3195) = 4.617$, $p = 0.004$; Argentina: $F(3311) = 0.382$, $p = 0.766$). In the Argentinian sample, neither years of occupational experience nor gender and age proved to be significant predictors, and the explanatory power was rather low (adj. $R^2 = -0.006$). In the first German model (adj. $R^2 = 0.052$), gender ($\beta = -0.219$, $p = 0.002$) and years of experience ($\beta = 0.198$, $p = 0.040$) proved to be significant predictors of self-efficacy; whereas, age did not play a role.

In the second model, internal and external reasons for successful online teaching were integrated as dummy variables while controlling for personal characteristics. A significant increase in the explained variance emerged for the German sample ($F(7191) = 6.139$, $p < 0.001$) as well as for the Argentinian sample ($F(7307) = 6.081$, $p < 0.001$). In the German model (adj. $R^2 = 0.154$), institutional support ($\beta = 0.192$, $p = 0.005$) as well as own technological skills ($\beta = 0.194$, $p = 0.004$) were relevant for the university teachers' self-efficacy; significant effects were also shown for the controlled personal characteristics, gender and years of experience ($p < 0.05$). Moreover, the sense of emergency proved to be predictive of German university teachers' self-efficacy ($\beta = -.184$, $p = 0.006$). In the Argentinian model (adj. $R^2 = 0.102$), we likewise found a positive effect on the self-efficacy for own technological skills ($\beta = 0.228$, $p < 0.001$). Further, addressing students' problems was a significant predictor of self-efficacy ($\beta = 0.193$, $p = 0.001$).

In the final regression model (Germany: $F(10,188) = 4552$, $p < 0.001$; Argentina: $F(10,304) = 6077$, $p < 0.001$), support needs for future crisis situations were integrated, and a significant increase in the explained variance emerged for the Argentinian sample only. In the German model (adj. $R^2 = 0.152$), none of the added variables were predictive for the self-efficacy of university teachers; whereas, in the Argentinian model (adj.

$R^2 = 0.139$) the support needs compensation ($\beta = 0.110$, $p = 0.044$) and more time to prepare ($\beta = -0.180$, $p = 0.003$) were significant predictors of self-efficacy.

In both final models, institutional support (external reason for success) and own technological skills (internal reasons for success) were predictors of self-efficacy in online teaching (see Table 2). In addition, gender and years of experience were predictive of self-efficacy in German university teachers; whereas, in the Argentinian data personal characteristics had no influence. Further analyses illustrate, that female teachers from Germany ($M = 3.14$; $SD = 0.41$) in particular rate themselves as competent and differ significantly from the male respondents in their self-efficacy ($M = 2.96$; $SD = 0.53$; $t$ (172,034) = 2.938, $p = 0.004$, Cohen's $d = 0.391$). In the Argentinian survey, however, this gender difference is not evident ($t(426) = 0.173$, $p = 0.862$, Cohen's $d = 0.017$). Moreover, the sense of emergency proved to be predictive of German university teachers' self-efficacy; whereas, for Argentinian university teachers' self-efficacy, addressing students' problems, as well as the support needs compensation and more time to prepare were relevant.

**Table 2.** Final regression models—dependent variable self-efficacy beliefs concerning online teaching.

| | Germany (n = 199) | | Argentina (n = 315) | |
|---|---|---|---|---|
| | *B* | β | *B* | β |
| personal characteristics | | | | |
| age | −0.003 | −0.009 | 0.023 | 0.057 |
| gender | −0.174 | −0.185 ** | 0.010 | 0.012 |
| years of experience | 0.199 | 0.217 * | 0.004 | 0.004 |
| competence and support | | | | |
| own technological skills | 0.183 | 0.196 ** | 0.223 | 0.216 *** |
| Addressing students' problems | 0.031 | 0.027 | 0.305 | 0.216 *** |
| institutional support | 0.123 | 0.214 ** | 0.074 | 0.129 * |
| sense of emergency | −0.178 | −0.183 ** | −0.049 | −0.056 |
| support needs | | | | |
| more time to prepare | −0.110 | −0.104 | −0.269 | −0.180 ** |
| professional support | −0.009 | −0.009 | −0.078 | −0.066 |
| compensation | 0.080 | 0.087 | 0.096 | 0.110 * |
| *F* | 4.552, $p < 0.001$ | | 6.077, $p < 0.001$ | |
| adjusted $R^2$ | 0.152 | | 0.139 | |

Notes. * $p < 0.05$, ** $p < 0.01$, *** $p < 0.001$

## 5. Discussion

Comparisons between diverse contexts concerning the perception of the pandemic-related ERT and the concept of self-efficacy beliefs are still scarce. Therefore, the present article focuses on the perception of the transition to online teaching during the first lockdown, and on the relevance of individual and institutional predictors for self-efficacy beliefs in online teaching, comparing the perspectives of university teachers from German with those from Argentinian universities in the context of the CRTS Study.

Results reveal a significant change in the use of digital tools by university teachers in Argentina and Germany when comparing the time before and during the first lockdown, especially for online lessons, discussion groups and self-produced videos. Although the transition to online teaching was a demanding phase with ambivalent experiences reported by university teachers, the overall picture of the presented analysis reveals a mostly successful implementation of online teaching by university teachers in both countries, providing interesting insights and valuable information for developing digital university teaching. These findings are in line with previous analysis in the respective countries, as well as analysis from the CRTS study in other European samples [5,6,27]. When comparing our results with former studies on online teaching in higher education [95], we can see a significant shift towards a higher usage levels of interactive online teaching methods. Concerning the success in online teaching with Zoom or other web conferencing systems,

university teachers from Argentina and Germany mostly consider themselves successful. When comparing both groups, the evaluation of this new experience is slightly more positive in the Argentinian sample than in the German sample. Further differences become apparent when looking at the use of digital tools: Argentinian university teachers report realizing a higher level of usage of online tools before the ERT than their German peers. Furthermore, Argentinian university teachers show higher ratings of their self-efficacy beliefs when compared with their German colleagues. The higher level of self-efficacy beliefs in online teaching might be due to the higher level of digital tool experience of the Argentinian university teachers before ERT; hence, the differences can be interpreted as differences in prior digital experience between the samples, which would be in line with self-efficacy research and actual findings in the context of ERT studies [57]. Differences in the level of ratings could also be a result of differing item interpretations between the Argentinian and German university teachers, which might have had an impact on their rating when confronted with the presented items [67].

The results of the regression analysis brought about further insights in understanding the connection between self-efficacy beliefs and individual and institutional factors. The findings support the idea of a connection between perceived competency in the use of technological tools in teaching and self-efficacy beliefs in online teaching [37,96]. The individual knowledge concerning technological tools is a relevant predictor for efficient online teaching [46,58]. In addition to self-reported ratings on technological competency, institutional support seems to be of general relevance for self-efficacy beliefs for both German and Argentinian university teachers. We see the general relevance of institutional support for digital teaching in both samples, meaning that universities should provide helpful infrastructure in terms of technologies, training measures and manuals to support their staff in mastering digital tools and digital didactics [57,63]. This finding underlines findings from various previous studies [6,50,58]. Furthermore, some kind of compensation or positive reinforcement for the implementation of digital teaching might be additionally motivating, as results from the Argentinian sample suggest.

The regression models for the prediction of self-efficacy beliefs in online teaching also hint at some slight differences between the samples. While for the Argentinian university teachers, the addressing of students' problems seems to be relevant for their self-efficacy beliefs, this is not the case within the German sample. This correlation between teacher-student relationship and self-efficacy beliefs in online teaching for Argentinian university teachers might hint at a specific relevance of the teacher-student relationship when compared with the German sample. In the German sample, we can see a negative correlation between the sense of emergency in the situation and a high perception of self-efficacy in online teaching. Those German colleagues who experienced a strong sense of emergency in the context of ERT reported lower self-efficacy beliefs in online teaching; in the Argentinian sample we do not find this correlation. This finding could hint at ERT as being a stressful event, especially for those university teachers who do not feel well-prepared for the ERT situation. As mentioned before, we found a higher rate of prior experience in online teaching and a more positive perception of the ERT situation in the Argentinian sample when compared with the German sample. Hence, the sense of stress might have been higher for the German university teachers in general; those who did not feel well prepared felt more distressed and reported lower self-efficacy beliefs. However, at the same time, this may also indicate implicit differences in attitudes towards digital tools in general (which also go hand-in-hand with the respective usage behaviors), which we cannot capture based on our data, but which need to be examined more closely in further studies.

Furthermore, we can see in the Argentinian sample that those respondents who reported needing more preparation time and who declared a need for compensation also reported lower self-efficacy beliefs in online teaching. Interestingly, this correlation is not significant for the German sample. German university teachers might not see the need for more preparation time as being relevantly connected with their conception of self-efficacy

in online teaching. This result might be related with different salary levels in both countries, with the Argentinian university teachers having lower salaries and generally having a higher need for receiving additional compensation than their German colleagues.

In line with findings from the US [62], and in contrast with other studies [60,61,97], which indicate higher self-efficacy in online teaching for male university teachers, our findings hint at gender differences concerning self-efficacy in online teaching in the German sample, with female university teachers reporting slightly higher levels of self-efficacy. In the Argentinian sample, we do not find these differences. Our results point at the heterogeneity of findings concerning online teaching and gender.

## 6. Conclusions

In conclusion, the relevance of individual experience and training as well as the relevance of supportive institutional conditions for self-efficacy beliefs can be noted for both samples. For future development of digital teaching in higher education, the promotion of support offers seems continuously relevant. Didactic offers should be made to support the potential use of digital tools, to foster experience, and develop positive attitudes. Furthermore, the expansion of technological infrastructure is a relevant precondition which has to be addressed continuously. Technology skills from both students and university teachers will have to be fostered in order to achieve digital transformation in higher education [95] and motivational aspects need to be addressed [44].

Still, our data leaves some open questions: How can we understand the differences in the perception of ERT between the samples? From a cross-cultural psychology perspective, one could argue, that Argentinians might generally be more prepared for the coping with ambivalent and unknown situations than Germans [98]. However, from the perspective of our study, we do not have data on differences in cultural values or coping strategies. From the point of view of data available in our study, we can see that Argentinian university teachers report more experience in online teaching prior to the ERT. This certain leading edge might have had a positive impact on their ERT experience. The Argentinian colleagues might have had more experience in digital teaching and hence were better prepared for online teaching in ERT. Furthermore, they reported receiving more support for their teaching, which in sum might make them experience this new situation as less stressful than their German peers. A further surprising result in the German sample is that female university lecturers reported higher self-efficacy beliefs than their male German peers; this is not the case for the Argentinian respondents. How can we explain this surprising result? We have more female participants in the German sample than in the Argentinian one. We might argue that self-selection processes might have played a relevant role; maybe more self-confident female university teachers participated in the German study, and maybe those who took the time to participate were more self-confident than their male peers. As the German sample is slightly younger, the results might also point at generational differences in online teaching self-efficacy beliefs.

The presented results should be interpreted carefully, as the study has several limitations. The most relevant limitation is the different sample sizes and sample compositions from both countries. Due to the voluntary nature of participation in both samples, self-selection processes could have led to a biased sample. It is possible that mainly lecturers with a generally positive attitude towards the digitalization of studies and teaching took part in our study. Overall, the present samples cannot be assumed to be representative of lecturers in Germany and Argentina. Some of the scales did not reveal high internal reliability, which also limited the opportunities for analysis. Furthermore, causal conclusions cannot be drawn from the cross-sectional design; it only allows an exploratory analysis of correlations within the data. Nevertheless, the results show interesting differences and similarities between the two contexts under research. Concerning the explanation for online self-efficacy, the regression models only explain 13–15% of the variance, hence there is a need to integrate further relevant variables into the model; for example, prior experience with digital tools. For further research, it would be helpful to gather larger samples and to

integrate longitudinal and qualitative perspectives into the design to learn more about the underlying beliefs and processes in both contexts.

**Author Contributions:** Conceptualization, K.G. and C.L.; Methodology, K.G. and K.N.; Formal analysis, K.N.; Investigation, K.G., C.L. and U.C.; Data curation, K.N.; Writing—original draft, K.G. and K.N.; Writing—review and editing, K.G., K.N., C.L. and U.C.; Supervision, K.G.; Project administration, K.G., C.L. and U.C. All authors have read and agreed to the published version of the manuscript.

**Funding:** This research received no external funding.

**Institutional Review Board Statement:** This research study was conducted in accordance with the declaration of Helsinki and approved by the data protection officer of the University of Duisburg-Essen (Germany).

**Informed Consent Statement:** Informed consent was obtained from all subjects involved in the study.

**Data Availability Statement:** Data available on request due to restrictions eg privacy or ethical The data presented in this study are available on request from the corresponding author.

**Conflicts of Interest:** The authors declare no conflict of interest.

## Appendix A

**Table A1.** Survey items and answer option.

| Survey Items | Answer Options |
|---|---|
| To what extent did you use digital tools in teaching before the coronavirus-related teaching situation (CRTS)?<br>To what extent do you use digital tools in teaching during the coronavirus-related teaching situation (CRTS)?<br><br>- LMS (learning management system) platform for syllabus and bibliography (e.g., Moodle, Adam, Ilias, Olat, Blackboard)<br>- LMS platform for discussion groups<br>- Presentations (e.g., PowerPoint, voice-recorded presentations)<br>- Selected videos from YouTube or others<br>- Self-produced videos<br>- Online lessons through Zoom or other tools | 1 = not at all<br>2 = to a small extent<br>3 = to a moderate extent<br>4 = to a large extent |
| Based on your general teaching experience in online teaching in particular, how would you describe your teaching experience during the CRTS?<br><br>- A very positive and inspiring experience<br>- A mostly positive and reassuring experience<br>- Nothing special, just as in conventional teaching<br>- A complex experience; it requires investment beyond what is expected<br>- A frustrating experience; it requires investment beyond what is expected<br>- An overwhelming experience; I hope this ends soon | 0 = not selected<br>1 = selected |
| In order to teach online, you had to learn to use advanced web conferencing systems, such as Zoom, or others. How was the preparation process? | 1 = very difficult<br>2 = difficult<br>3 = reasonable<br>4 = simple<br>5 = very simple |

**Table A1.** *Cont.*

| Survey Items | Answer Options |
|---|---|
| How do you consider your first lessons using Zoom or other web conferencing system? | 1 = unsuccessful<br>2 = could be better<br>3 = reasonable<br>4 = successful<br>5 = very successful |
| To what extent would you expect support of your institution in time of a future crisis?<br>- More time to prepare for online teaching<br>- Support from professionals to assist you to teach courses online<br>- To provide compensation for preparing courses online | 1 = not at all<br>2 = to a small extent<br>3 = to a moderate extent<br>4 = to a large extent |
| I feel confident that I am able to:<br>- meet my students' expectations despite the current crisis.<br>- use digital tools as a means to maintain the same quality of teaching.<br>- be flexible in my teaching and alter my plans.<br>- determine how difficult newly introduced technology will be for my students to use.<br>- select the most efficient digital tools for teaching in this situation.<br>- select the appropriate online method to effectively convey course content used before the crisis.<br>- adapt ne newly designed online courses to the needs of my students.<br>- accurately assess the academic progress of my students. | 1 = not at all<br>2 = somewhat disagree<br>3 = somewhat agree<br>4 = completely agree |
| To what extent do you think that the success of your online teaching in the current context is due to any of the following reasons?<br>- My own technological skills<br>- Acknowledging and addressing issues faced by the students<br>- Sense of emergency<br>- Technological pedagogical support of the academic institution<br>- Existing online tutorials<br>- Support and encouragement of administration/senior management of the institution | 1 = not at all<br>2 = to a small extent<br>3 = to a moderate extent<br>4 = to a large extent |

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
