# Peer review of "Self-Efficacy in Online Teaching during the Immediate Transition from Conventional to Online Teaching in German and Argentinian Universities—The Relevance of Institutional Support and Individual Characteristics"

_education, doi:10.3390/educsci13010076_

Round 1

Reviewer 1 Report

Thank you very much for the opportunity to review your work. Overall, I really enjoyed reading this.

Lockdown is one word – please amend as necessary in the paper.

Please include the total number of participants in the abstract just as a (n = 728) after the word Argentina on line 13.

You mention ERT in your abstract – I know it’s emergency remote teaching, but others may not. Perhaps consider removing it from the abstract.

Page 2, line 53 – effectivity isn’t really a word – effectiveness would be better.

Whilst this is quite well-written, I would suggest there are a few tense issues here and there.

Page 2, line 70 – are you able to cite a review perhaps that shows that the experiences of uni teachers has only been addressed in a few studies? Perhaps you could cite Bond et al. (2021) here, which you have included in your reference list.

Just wondering whether the numbers you’re citing need to be in numerical order? For some reason my brain doesn’t like reading the numbers when they’re out of order, but maybe that’s just me?

Page 2, line 80 – this first sentence is a little confusing. Could you please revise it? There is also a space after [36] that should be deleted.

Page 3, line 102 – you need to have literature to support the claim that self-efficacy in online teaching was scarce.

Page 3 has a massive paragraph – this must be revised so that multiple paragraphs exist.

Page 3, line 142 – you’ve missed a number for authors, I think?

I would almost have comparative perspectives on ERT as section 1.3 and then have something like ‘The experience of ERT in Germany and Argentina’ as section 2. Just a thought.

Double quotation marks should only be used if you’re directly quoting (line 215) – otherwise, use single ones.

Page 5, line 227 – need a number instead of Riedel?

Page 5, line 236 – is there a full stop missing?

Page 5, line 239 – the citing literature should go after 2020.

Page 5, line 244 – the full stop in 25,000 should be a comma.

Page 6, line 284 – percentages should have a full stop rather than a coma – 99.5%. Please change this throughout the paper.

Your introduction is really very long – I wonder whether a slightly more succinct introduction would be better? Focus mostly on Germany and Argentina.

Page 7, line 333 – The present data is derived

Page 7, line 336 – Spring should have a capital

Page 8, line 364 – consider changing cf. to see. I would also consider removing the space between the number and percentage sign, e.g. 31.4% instead of 31.4 %

Page 9, line 407 and page 10, line 443– you still have the author names there.

Is the survey available somewhere in its entirety? Please attach as an appendix or attach a link to where it can be found (e.g. OSF).

You might be interested in this German study to compare your results to: Bond, M., Marin, V., Dolch, C., Bedenlier, S., & Zawacki-Richter, O. (2018) Digital transformation in German higher education: student and teacher perceptions and usage of digital media. International Journal of Educational Technology in Higher Education, 15(1), 1-20. https://doi.org/10.1186/s41239-018-0130-1

I’m just wondering whether you asked exactly how long each university was in lockdown/ERT for?

Page 12, line 511 – do you mean they expect monetary compensation over and above their salaries for preparing online resources? This might need slightly more clarification here (aside from what is in the discussion section).

Page 14, line 574 – occupation not occupational

Page 15, line 612 – I think you mean diverse rather than ambivalent

Page 16, line 684 – please specify which studies do and which studies don’t agree with your findings there.

Please consider adding a separate conclusion section, or call section 4 Discussion and Conclusion.

Page 16, line 689 – In conclusion, instead of concluding,

Page 16, line 693 – I know digital media and media are very Germany specific terms, but they mean different things in the English-speaking world. Please consider changing this to something like digital tools.

Page 17, line 702 – report not reporting

Author Response

Dear editors and reviewers,

Thank you very much for the review of our contribution. Your feedback was very helpful to improve the manuscript. We have revised the paper based on your comments and hope to have optimized it in accordance with your remarks.

With kind regards

Authors

Comment

Response

General feedback

1

Thank you very much for the opportunity to review your work. Overall, I really enjoyed reading this.

Thank you for the appreciation of our work!

Abstract

1

Lockdown is one word – please amend as necessary in the paper.

We checked the spelling of ‘lockdown’.

Please include the total number of participants in the abstract just as a (n = 728) after the word Argentina on line 13.

We added the number of participants.

You mention ERT in your abstract – I know it’s emergency remote teaching, but others may not. Perhaps consider removing it from the abstract.

ERT was removed from the abstract.

Introduction

1

Whilst this is quite well-written, I would suggest there are a few tense issues here and there.

We checked and corrected the use of tenses on the introduction.

1

Just wondering whether the numbers you’re citing need to be in numerical order? For some reason my brain doesn’t like reading the numbers when they’re out of order, but maybe that’s just me?

We agree and revised the numerical order in our citations.

1

Page 2, line 53 – effectivity isn’t really a word – effectiveness would be better.

Page 2, line 70 – are you able to cite a review perhaps that shows that the experiences of uni teachers has only been addressed in a few studies? Perhaps you could cite Bond et al. (2021) here, which you have included in your reference list.

Page 2, line 80 – this first sentence is a little confusing. Could you please revise it? There is also a space after [36] that should be deleted.

The spelling was corrected.

Thank you for this advice, we now added this citation.

We revised this sentence.

1

Page 3, line 102 – you need to have literature to support the claim that self-efficacy in online teaching was scarce.

Page 3 has a massive paragraph – this must be revised so that multiple paragraphs exist. Page 3, line 142 – you’ve missed a number for authors, I think?

We changed this paragraph and deleted  this claim.

We agree and restructured this chapter.

Yes, thank you.

1

I would almost have comparative perspectives on ERT as section 1.3 and then have something like ‘The experience of ERT in Germany and Argentina’ as section 2. Just a thought.

Thank you for idea, we decided to stay with the original order of sections

1

Double quotation marks should only be used if you’re directly quoting (line 215) – otherwise, use single ones.

Thank you for this helpful advice.

1

Page 5, line 227 – need a number instead of Riedel?

Page 5, line 236 – is there a full stop missing?

Page 5, line 239 – the citing literature should go after 2020.

Page 5, line 244 – the full stop in 25,000 should be a comma.

We integrated the author of the study as well as the citation number.

Yes, we added the full stop.

We agree and added recent literature.

Yes, thank you.

1

Page 6, line 284 – percentages should have a full stop rather than a coma – 99.5%. Please change this throughout the paper.

We corrected the presentation of percentages.

1

Your introduction is really very long – I wonder whether a slightly more succinct introduction would be better? Focus mostly on Germany and Argentina.

We shortened the introduction and tried to get more focused.  

Method

1

Page 7, line 333 – The present data is derived

Page 7, line 336 – Spring should have a capital

Thank you, we corrected those mistakes.

1

Page 8, line 364 – consider changing cf. to see. I would also consider removing the space between the number and percentage sign, e.g. 31.4% instead of 31.4 %

Thank you, we revised this aspect.

1

Page 9, line 407 and page 10, line 443– you still have the author names there.

Yes, thank you for this helpful advice. We deleted the authors’ names.

1

Is the survey available somewhere in its entirety? Please attach as an appendix or attach a link to where it can be found (e.g. OSF).

We attached the survey as an appendix.

1

I’m just wondering whether you asked exactly how long each university was in lockdown/ERT for?

This aspect was no part of our survey. Therefore, we unfortunately have no information about how long each university was in lockdown.

Results

1

Page 12, line 511 – do you mean they expect monetary compensation over and above their salaries for preparing online resources? This might need slightly more clarification here (aside from what is in the discussion section).

We agree and added some more information about this item.

1

Page 14, line 574 – occupation not occupational

Thanks, we corrected this mistake.

Discussion

1

You might be interested in this German study to compare your results to: Bond, M., Marin, V., Dolch, C., Bedenlier, S., & Zawacki-Richter, O. (2018) Digital transformation in German higher education: student and teacher perceptions and usage of digital media. International Journal of Educational Technology in Higher Education, 15(1), 1-20.

We integrated a comparative perspective towards the results from Bond et al., 2018 into our discussion

1

Page 15, line 612 – I think you mean diverse rather than ambivalent

Was revised in the text

1

Page 16, line 684 – please specify which studies do and which studies don’t agree with your findings there.

Please consider adding a separate conclusion section, or call section 4 Discussion and Conclusion.

Page 16, line 689 – In conclusion, instead of concluding,

Page 16, line 693 – I know digital media and media are very Germany specific terms, but they mean different things in the English-speaking world. Please consider changing this to something like digital tools.

We specified the content of the studies and deleated irrelevant result.

Since reviewer 2 also made this suggestion, we have divided the chapter into Discussion and Conclusion.

Revised in the text.

Thanks for this helpful remark. We changed the terms into ‘digital tools’.

1

Page 17, line 702 – report not reporting

We corrected this mistake.

Reviewer 2 Report

Dear authors,

I had thoroughly read your manuscript. Despite the fact that this manuscript contained a sufficient number of citations, I would suggest the following enhancements: I believe that a portion of the manuscript and a number of its fundamental premise lack recent and appropriate references.

For instance, page 2, lines 63 to 69 are intriguing because they discuss the impact of lockdown on students, particularly when the authors are able to pinpoint the gender-specific effects of lockdown. However, the discussion is limited because it focuses solely on mental health, activity, and satisfaction. As motivation can have an effect on an individual's sense of self-efficacy, perhaps the authors could elaborate further on the gender-based differences in student motivation. It would be wonderful if the authors could also shed light on gender differences in online learning acceptance.

Consequently, based on my search in Scopus, I believe that these two articles would be useful to the author's manuscript for the stated purpose, as they discuss the effect of gender on self-efficacy and are recently published.

Rosli, M.S., Saleh, N.S. Technology enhanced learning acceptance among university students during Covid-19: Integrating the full spectrum of Self-Determination Theory and self-efficacy into the Technology Acceptance Model. Curr Psychol (2022). https://doi.org/10.1007/s12144-022-02996-1

Link: https://rdcu.be/cZYOs

Sigalit Warshawski, Academic self-efficacy, resilience and social support among first-year Israeli nursing students learning in online environments during COVID-19 pandemic, Nurse Education Today, Volume 110, 2022, 105267, https://doi.org/10.1016/j.nedt.2022.105267.

Link: https://www.sciencedirect.com/science/article/pii/S026069172200003X

Page 2, paragraph 2: This has relatively poor continuity with the preceding paragraph, which discussed student stress. Perhaps the authors could revisit this and strengthen the connection between these two statements.

Page 4, line 183 – 186: Please provide a citation for the CRTS study. In order for the readers to gain a deeper understanding of the instrument used and the background and foundation of the study.

Line 344 – 347: How about the ethical implications of mentioning the institution of the samples in this research? It is stated here that the German study was conducted at the University of Duisburg-Essen. In the section on participants, however, it was stated that the research was conducted at numerous German universities, with the University of Duisburg-Essen being the primary contributor. Kindly check these two concerns.

I would recommend that the authors divide the discussion and conclusion into two distinct subsections.

I hope that my comments will aid the authors in improving the quality of their manuscripts. Best wishes, and hopefully the manuscripts will receive a significant number of citations.

Author Response

Dear editors and reviewers,

Thank you very much for the review of our contribution. Your feedback was very helpful to improve the manuscript. We have revised the paper based on your comments and hope to have optimized it in accordance with your remarks (see table enclosed).

With kind regards

Authors
